# An analysis of behavioral characteristics and enrollment year variability in 47,444 dogs entering the Dog Aging Project from 2020 to 2023

Yuhuan Li[1◉], Courtney L. Sexton[2◉*], DAP Consortium[¶], Annette Fitzpatrick[3], Audrey Ruple[2]

**1** Department of Biostatistics, School of Public Health, University of Washington, Seattle, Washington, United States of America, **2** Department of Population Health Sciences, Virginia-Maryland College of Veterinary Medicine, Virginia Polytechnic Institute and State University, Blacksburg, Virginia, United States of America, **3** Department of Epidemiology, School of Public Health, University of Washington, Seattle, Washington, United States of America

¶ Membership of the DAP Consortium is provided in the Acknowledgements.
◉ These authors contributed equally to this work.
* sextonc@vt.edu

## Abstract

Understanding dog behavior, especially in the context of the human social environment, is critical to maintaining positive human-dog interactions and relationships. Furthermore, behavior can be an important indicator of health and welfare in companion dogs. Behavioral change can signal transitions in life stages, alert caretakers to potential illnesses or injuries, and is an important factor in understanding and measuring stress. In order to take advantage of behavioral change as a biomarker, however, we must first have a behavioral baseline to assess. Thus, using owner-reported data from dogs enrolled in the Dog Aging Project (DAP) from 2020–2023, our aim was to establish baseline behavioral measures for 47,444 dogs, with the goal of using these measures in future research investigating behavioral change in dogs and short- and long-term health outcomes. Given that the data collection period spanned the 2019 coronavirus disease (COVID-19) lockdown period and its immediate aftermath, a secondary aim of this study was to evaluate whether year of project entry impacted average reported behavior scores in dogs and to investigate additional variables that may influence observed differences. In our analyses of cohort baseline and year-over-year changes among four composite behavior domains — Fear, Attention/Excitability, Aggression, and Trainability — we find that time (year of enrollment) had the highest influence on Trainability, wherein dogs enrolled in all three years after 2020 (2021–2023) had lower average reported scores than dogs enrolled in 2020. Several other variables, including breed, life stage, sex, spay/neuter status, size, primary residence, and primary activities, have positive and negative statistical associations with mean behavioral scores in all four domains.

**Data availability statement:** This research is based on publicly available data collected by the Dog Aging Project. These data are housed on the Terra platform at the Broad Institute of MIT and Harvard. Further information, including the data access request form, is available here: https://dogagingproject.org/data-access.

**Funding:** This research is based on publicly available data collected by the Dog Aging Project, under U19 grant AG057377 (PI: Daniel Promislow) from the National Institute on Aging, a part of the National Institutes of Health, and by additional grants and private donations, including generous support from the Glenn Foundation for Medical Research, the Tiny Foundation Fund at Myriad Canada, and the WoodNext Foundation.

**Competing interests:** The authors have declared that no competing interests exist.

## Introduction

While early exposures and environment influence how behaviors emerge in dogs, many behavioral phenotypes can be linked phylogenetically to specific breeds and breed groups [1–5]. In addition to and beyond breed-specific traits, behavior, broadly, is inextricably linked to clinical outcomes and welfare for domestic dogs. In everyday contexts, behaviors are an, effective means to orient people to a dog's emotional state at a given time or in a given scenario, and to help predict how a dog may respond to various stimuli [6].

When considering aspects of behavior related to health, behavior changes can be both triggers and indicators of emergent physical or cognitive issues as well as declining health over time [7–10]. Dogs that are in acute or chronic physical pain often display sudden shifts in mood or responses to otherwise "normal" stimuli, as well as gradual changes in typical behavior over time, depending on the nature of the ailment [11–13].

In regard to cognitive and emotional states, reported changes and disruptions in dogs' routines such as amount of time left alone and even Daylight Saving Time have led to observable behavior changes in animals [14,15]. Emerging research into coronavirus disease 19 (COVID-19) pandemic-related changes to dogs' households and daily activities also offer a unique point-in-time record of this effect, such as in one study where dogs previously inclined to separation-related behavioral issues were more likely to develop more problems during lockdown [16], and another where owners conversely reported more positive changes in their dogs [17]. Furthermore, beyond shifts in routines, dogs who have experienced trauma, especially in early developmental stages, are prone to developing new and/or altered behaviors, including fear-aggression, avoidance, self-licking, hyperactivity, attachment and attention-seeking, anxiety, repetitive vocalizing, etc. [18–21]. Fortunately, there are a number of behavior modification tools and strategies that can ameliorate undesirable behaviors and improve dogs' quality of life, if implemented correctly and at the right time (preemptively and/or once a "problem" behavior has emerged) [22–25].

Having an awareness of a dog's baseline behavioral profile is a crucial component of being able to detect, record, and, when necessary, address behavioral changes in both the short and long term. From a research perspective, establishing this baseline is also important in order to monitor the relationship between behavior and physiological changes over dogs' lifetimes. Furthermore, in addition to characterizing behavior in individual dogs, establishing group-level behavioral profiles is important for tracking changes in a population over time, which could also have important translational applicability.

Dog behavior can be reliably assessed using the Canine Behavioral Assessment and Research Questionnaire (C-BARQ), an extensive and widely-used survey tool that captures dogs' behavioral profiles via responses to questions presented on a 5-point ranking scale [26]. Survey items cover 14 domains of dog behavior: stranger-directed aggression, owner-directed aggression, dog-directed aggression, stranger-directed fear, non-social fear, dog-directed fear, dog rivalry, separation-related behavior, attachment/attention-seeking behavior, trainability, chasing, excitability,

touch sensitivity and energy level (Table 1). Although C-BARQ scores are generated by information from dog owners and thus have a certain degree of inherent subjectivity, the instrument has proven useful for studying behavior in large numbers of dogs. The survey has derived accurate measures of behavioral profiles in various contexts, including as a standard used by guide dog organizations in taking reports from puppy raisers on the behaviors and habits of future service dogs [27–30].

The Dog Aging Project (DAP), a long-term, longitudinal study of life histories and aging in companion dogs in the U.S., utilizes a shortened version of the C-BARQ (~40 question items instead of ~100) [31] as a component of the Health and Life Experience Survey (HLES)— the comprehensive form that owner participants complete when enrolling their dog in the study. This shortened version of the C-BARQ was validated at the domain level against the long version by DAP researchers [32]. Via the HLES and subsequent annual follow-up surveys, the DAP collects owner-reported lifestyle, health, and behavioral data on more than 40,000 dogs enrolled in the study [33].

As such, the DAP sample provides a robust opportunity to establish baseline behaviors for a large cohort of dogs that can ultimately be re-evaluated to identify changes in behavior both in individual dogs over time and at the cohort and population level. Here, we evaluate the C-BARQ domains for DAP dogs at the time of entry into the study, examining dogs who entered the study each year from 2020–2023. Our aim is twofold: 1) to describe behavioral data for specific characteristics for cohorts of DAP dogs at their time of enrollment; and 2) to examine behavioral data among cohorts of dogs enrolled each year over the lifetime of the project for similarities and differences and investigate additional variables that may impact observed differences, especially in light of the fact that the data collection period spanned the COVID-19 lockdown and its immediate aftermath. This is the first group-level publication of behavioral characteristics of a long-term longitudinal study in which dogs will continue to age. With these group-level measures established, we'll be able to track changes for the whole population over time, in addition to individual data acquired through the HLES.

## Methods

### Ethics statement

The University of Washington IRB deemed that recruitment of dogs through their owners for the Dog Aging Project is research that qualifies for Category 2 human subjects exempt status (IRB ID no. 5988, effective 30 October 2018). No interactions between researchers and privately owned dogs occurred; therefore, IACUC oversight was not required.

**Table 1. Behavioral domains included in the Canine Behavioral Assessment and Research Questionnaire (C-BARQ).**

| Behavioral Domain | |
|---|---|
| 1 | Stranger-directed Aggression |
| 2 | Owner-directed Aggression |
| 3 | Stranger-directed Fear |
| 4 | Non-social Fear |
| 5 | Dog-directed Aggression |
| 6 | Dog-directed Fear |
| 7 | Dog Rivalry |
| 8 | Separation-related Behavior |
| 9 | Attachment or Attention-seeking Behavior |
| 10 | Trainability |
| 11 | Chasing |
| 12 | Excitability |
| 13 | Touch Sensitivity |
| 14 | Energy Level |

## Data collection

Data used in this study were collected as part of the DAP, an open data project conducted in the population of companion dogs in the United States. These data are available through completion of a data access process (dogagingproject.org/open_data_access). To enroll a dog in the DAP, dog owners complete a short nomination survey on the DAP website (www.dogagingproject.org) and set up a secure personal research portal. This portal utilizes the REDCap electronic data capture system [34] to collect survey information from the owner throughout the duration of their participation in the DAP. The owner is led through an informed consent process. Once consent is obtained, owners complete the Health and Life Experience Survey (HLES), which collects extensive information about the home environment, activity and behavioral habits of the dog, and the dog's health history. Once enrolled, owners are asked to complete the Annual Follow-up Survey (AFUS) annually, as well as other voluntary surveys and activities throughout the year.

## Sample

Data used in this study were extracted from the 2023 DAP Curated Data Release, which includes records from 2020 through 2023. Over this time period, a total of 47,444 dogs were enrolled in the DAP and submitted complete HLES information including responses to the abbreviated C-BARQ survey. These records include the basic demographic characteristics of dogs, as well as their lifestyles, behaviors, and health conditions as reported by their owners.

## Statistical analyses

The C-BARQ produces a set of 14 behavioral domains, + a "Miscellaneous" domain, that are calculated based on specific areas of dog behaviors (Table 1). Acknowledging a high correlation among these domains, we used a principal component analysis (PCA) to reduce the number of outcomes and obtain key behavior areas represented in our data. A Kayser-Meyer-Olkin (KMO) test (an overall measure of sample adequacy) was > 0.5 (0.74), which indicated a sample size sufficient for PCA analysis, and Bartlett's Test of Sphericity confirmed suitability of PCA analysis ($p < .05$). The PCA was performed using orthogonal rotation with varimax option to derive optimal non-correlated components.

The correlation matrix of the standardized variables was examined to determine the number of components to retain based on eigenvalue and interpretability. Scree Plot analysis produced four principal components with eigenvalues >1 (above the simulated and resampled threshold). Given this strong indicator for meaningful signal, these four components were used in further evaluation. Based on clustering in the four principal components, the final key behavior areas, named as Fear, Attention/Excitability, Aggression, and Trainability, were used as outcomes in regression analyses.

We fit four multivariable linear regression models, one for each principal component, to estimate associations between canine characteristics and key behavior domains. Covariates included: age (in the form of "life stage"); sex and reproductive status (male/female; neutered/intact); size (weight class); breed type (single-breed or mixed-breed); primary activity (companion/pet, assistance or therapy, obedience, service, other); insured status (yes/no); and U.S. geographic region of residence (Midwest, Northeast, South, West) (S1 Table in S1 File):

$$Behavior = \beta_0 + \beta_1 * Year + \beta_2 * Breed + \beta_3 * LifeStage + \beta_4 * Sex + \beta_5$$

$$* Spay/NeuterStatus + \beta_6 * Size + \beta_7 * HealthInsuranceStatus + \beta_8$$

$$* U.S.RegionofResidence + \beta_9 * PrimaryActivities$$

We wanted to examine if dogs entering the study in different years demonstrated behavioral differences independent of changes over time, given that a puppy entering the study in 2020 could likely have different experiences than a puppy entering the study in 2022. To examine the effect of entry-year cohorts on each of the effects and covariates interaction terms for main effects and covariates included in final models were evaluated (S2 Table in S1 File):

$$\text{Behavior} = \beta_0 + \beta_1 * \text{Year} + \beta_2 * \text{Breed} + \beta_3 * (\text{Breed} \times \text{Year}) + \beta_4$$

$$* \text{LifeStage} + \beta_5 * (\text{LifeStage} \times \text{Year}) + \beta_6 * \text{Sex} + \beta_7$$

$$* (\text{Sex} \times \text{Year}) + \beta_8 * \text{Spay/NeuterStatus} + \beta_9$$

$$* (\text{Spay/NeuterStatus} \times \text{Year}) + \beta_{10} * \text{Size} + \beta_{11}$$

$$* (\text{Size} \times \text{Year}) + \beta_{12} * \text{HealthInsuranceStatus} + \beta_{13}$$

$$* (\text{HealthInsuranceStatus} \times \text{Year}) + \beta_{14}$$

$$* \text{U.S.RegionOfResidence} + \beta_{15} * (\text{U.S.RegionOfResidence} \times \text{Year}) + \beta_{16}$$

$$* \text{PrimaryActivities} + \beta_{17} * (\text{PrimaryActivities} \times \text{Year})$$

Finding no statistical significance or else negligible effects for interaction terms (S2 Table in S1 File), these were not included in the final analysis.

We referred to DAP standards for binning life stage and dog size. Life stage for dogs enrolled in DAP is determined by the 2019 AAHA Canine Life Stage Guidelines [35] as previously described [36]. Dog size as reported by owners is binned according to 20-pound (~9 kg) increments from <20 pounds to >100 pounds (<9 kg to >45 kg).

Given that household activities were expected to have changed dramatically due to the COVID-19 pandemic during the period of data collection, we adjusted for year and adjusted for additional demographic characteristics (age, sex, life stage, breed type, primary activity, and geographic region).

We described characteristics of the sample by enrollment year using means and standard deviations for continuous variables and percentages for categorical variables, and we compared the means of characteristics between enrollment years. We used ANOVA to compare the means of overall average behavioral domain scores across years.

We used R (version 4.4.2 [37], packages missMDA [38] and psych [39]) to run statistical analyses.

## Results

### Demographic information

Of the sample of 47,444 dogs included in our analyses (Table 2), half of the dogs (23,857) were reportedly single-breed dogs and the other half (23,587) were reported to be mixed-breed dogs. The sample was likewise evenly split between male and female dogs, with most dogs (88%) neutered/spayed. The majority of dogs (55%) in the sample were mature adults. Puppies comprised 5% of the total sample, young adult dogs 21%, and seniors 18%. Sixty-five percent of dogs were in weight classes up to 60 pounds (~27 kg), with 35% in classes inclusive of weights 61 pounds (~28 kg) or greater. Dogs lived in all U.S. regions, with the greatest percentage (35%) living in the West, followed by 30% in the South, 19% in the Midwest, and 16% in the Northeast. The majority of dogs (74%) were primarily companion animals, and most (79%) were not medically insured.

### Overall average behavioral score

We establish behavioral data at the time of enrollment for 47,444 dogs across 14 behavioral domains. Average behavioral scores differed among enrollment years in each domain (Table 3). In domain 1 (Stranger-directed Aggression), domain 5 (Dog-directed Aggression), domain 7 (Dog Rivalry) and domain 12 (Excitability), average scores were lower with each successive year of enrollment from 2020 to 2023. In domain 3 (Stranger-directed Fear) and domain 9 (Attachment or

**Table 2. Distribution of demographic characteristics for all dogs included in the analysis of behaviors. Dog Aging Project, 2020–2023.**

| DEMOGRAPHIC VARIABLES | | N of dogs/ YEAR of enrollment | | | | |
|---|---|---|---|---|---|---|
| | | **2020** | **2021** | **2022** | **2023** | **TOTAL** |
| **Breed** | Single | 13573 | 3007 | 5309 | 1968 | 23875 |
| | Mixed | 13864 | 2646 | 5070 | 2007 | 23587 |
| **Life Stage at Health and Life Experience Survey (HLES)** | Puppy | 590 | 537 | 1047 | 351 | 2525 |
| | Young Adult | 5074 | 1125 | 2948 | 944 | 10091 |
| | Mature Adult | 16227 | 3003 | 4952 | 2039 | 26221 |
| | Senior | 5543 | 977 | 1406 | 630 | 8556 |
| | Unknown | 3 | 11 | 26 | 11 | 51 |
| **Sex** | Male | 13771 | 2905 | 5235 | 1979 | 23890 |
| | Female | 13666 | 2748 | 5144 | 1996 | 23554 |
| **Spay/Neuter Status** | Not intact | 25320 | 4717 | 8507 | 3344 | 41888 |
| | Intact | 2117 | 936 | 1872 | 631 | 5556 |
| **Size (weight class)** | <20lbs (~9 kg) | 5544 | 1052 | 1994 | 874 | 9464 |
| | 21-40lbs (~10–18 kg) | 5156 | 1042 | 2172 | 752 | 9122 |
| | 41-60lbs (~19–27 kg) | 6858 | 1422 | 2744 | 1038 | 12062 |
| | 61-80lbs (~28–36 kg) | 6286 | 1227 | 2240 | 840 | 10593 |
| | 81-100lbs (~37–45 kg) | 2399 | 564 | 811 | 323 | 4097 |
| | >100lbs (>45kgs) | 1194 | 346 | 418 | 148 | 2106 |
| **U.S. Region of Residence** | Northeast | 4255 | 812 | 1875 | 748 | 7690 |
| | South | 7889 | 1762 | 3197 | 1266 | 14114 |
| | Midwest | 5743 | 982 | 1815 | 679 | 9219 |
| | West | 9549 | 2097 | 3491 | 1281 | 16418 |
| | Unknown | 1 | 0 | 1 | 1 | 3 |
| **Health Insurance Status** | Insured | 4869 | 1256 | 2813 | 1189 | 10127 |
| | Not insured | 22568 | 4397 | 7566 | 2786 | 37317 |
| **Primary Activities** | Companion | 26209 | 5321 | 9757 | 3750 | 35036 |
| | Assistance/therapy | 184 | 42 | 75 | 34 | 335 |
| | Obedience | 181 | 44 | 89 | 40 | 354 |
| | Service | 221 | 65 | 129 | 47 | 462 |
| | Other | 643 | 181 | 329 | 104 | 1257 |
| **TOTAL** | | **27437** | **5653** | **10379** | **3975** | **47444** |

Attention-seeking), average scores were higher with each successive year of enrollment from 2020 to 2023. For other domains, there were not monotone trends, rather instances of year-to-year increase, decrease, and/or stasis.

### Principal component analysis (PCA)

Four principal components explained 50% of the variability in dogs' behavior scores (Table 4): Behavioral domains with the highest loadings clustered in PC1 included Stranger-directed Aggression, Stranger-directed Fear, Non-social Fear, and Dog-directed Fear, resulting in a PC1 named "Fear". Behavioral domains with the highest loadings clustered in PC2 included Separation-related and Attachment or Attention-Seeking Behavior, Chasing, Excitability, and Energy level, resulting in a PC2 named "Attention". Behavioral domains with the highest loadings clustered in PC3 included Owner-directed Aggression and Dog Rivalry, resulting in a PC3 named "Aggression". Behavioral domains with the highest loadings clustered in PC4 included Trainability and Touch Sensitivity, resulting in a PC4 called "Trainability".

**Table 3. Mean C-BARQ behavior scores for each of four enrollment years, comprising a total of 47,444 dogs enrolled in the Dog Aging Project from 2020–2023.**

| | | Mean Score Per Enrollment Year | | | | Mean Across Years | p-value |
|---|---|---|---|---|---|---|---|
| **C-BARQ Domain** | | **2020 N = 27437** | **2021 N = 5653** | **2022 N = 10379** | **2023 N = 3975** | | |
| 1 | Stranger-directed Aggression | 0.824 | 0.775 | 0.762 | 0.731 | 0.773 | <0.001 |
| 2 | Owner-directed Aggression | 0.137 | 0.137 | 0.134 | 0.122 | 0.132 | 0.14 |
| 3 | Stranger-directed Fear | 0.652 | 0.659 | 0.673 | 0.674 | 0.664 | 0.187 |
| 4 | Non-social Fear | 1.044 | 1.019 | 1.013 | 1.019 | 1.024 | 0.002 |
| 5 | Dog-directed Aggression | 1.311 | 1.212 | 1.175 | 1.136 | 1.209 | <0.001 |
| 6 | Dog-directed Fear | 1.229 | 1.185 | 1.174 | 1.210 | 1.200 | <0.001 |
| 7 | Dog Rivalry | 0.595 | 0.583 | 0.558 | 0.540 | 0.569 | 0.001 |
| 8 | Separation-related Behavior | 0.628 | 0.701 | 0.709 | 0.685 | 0.681 | <0.001 |
| 9 | Attachment or Attention-seeking Behavior | 2.766 | 2.779 | 2.822 | 2.846 | 2.803 | <0.001 |
| 10 | Trainability | 2.560 | 2.569 | 2.590 | 2.587 | 2.577 | <0.001 |
| 11 | Chasing | 1.935 | 1.903 | 1.958 | 1.867 | 1.916 | <0.001 |
| 12 | Excitability | 2.273 | 2.221 | 2.207 | 2.168 | 2.217 | <0.001 |
| 13 | Touch Sensitivity | 1.045 | 1.037 | 1.076 | 1.090 | 1.062 | 0.004 |
| 14 | Energy Level | 1.453 | 1.541 | 1.614 | 1.580 | 1.547 | <0.001 |
| Misc. | Others | 0.644 | 0.673 | 0.698 | 0.680 | 0.673 | <0.001 |

**Table 4. Results of PCA. Principal Components 1-4 are linear combinations of the C-BARQ behavioral domains with loadings (i.e., how much they contributed to each domain) as their coefficients. Dog Aging Project, 2020–2023.**

| C-BARQ Domain | | PC1 | PC2 | PC3 | PC4 |
|---|---|---|---|---|---|
| 1 | Stranger-directed Aggression | 0.60 | 0.13 | 0.39 | 0.28 |
| 2 | Owner-directed Aggression | −0.01 | 0.08 | 0.73 | −0.22 |
| 3 | Stranger-directed Fear | 0.77 | 0.09 | −0.05 | −0.13 |
| 4 | Non-social Fear | 0.67 | 0.17 | −0.04 | −0.28 |
| 5 | Dog-directed Aggression | 0.54 | 0.01 | 0.50 | 0.36 |
| 6 | Dog-directed Fear | 0.75 | 0.02 | 0.10 | 0.00 |
| 7 | Dog Rivalry | 0.10 | 0.05 | 0.76 | −0.06 |
| 8 | Separation-related Behavior | 0.14 | 0.56 | 0.08 | −0.36 |
| 9 | Attachment or Attention Seeking Behavior | 0.04 | 0.48 | −0.11 | −0.07 |
| 10 | Trainability | −0.04 | 0.15 | −0.12 | 0.60 |
| 11 | Chasing | 0.10 | 0.47 | 0.22 | 0.32 |
| 12 | Excitability | 0.10 | 0.50 | 0.11 | 0.19 |
| 13 | Touch Sensitivity | 0.33 | 0.17 | 0.17 | −0.43 |
| 14 | Energy level | −0.04 | 0.75 | −0.05 | 0.18 |
| | Miscellaneous | 0.14 | 0.68 | 0.22 | −0.13 |

## Regression analyses

We found no statistical significance or else negligible effects for interaction terms. In particular, the interaction between dog life stage and year of entry into the study (time of completion of HLES) did not impact behavioral score averages across domains (S1 and S2 Tables in S1 File).

**Fear.** With 2020 as a baseline (reference for year variable), our data showed no significant change in dogs' behaviors in the Fear (PC1) domain across enrollment years 2020–2023.

Analyses of additional variables of interest revealed significant relationships with reported Fear behaviors. Mixed-breed dogs had a higher mean behavior score in Fear (were more fearful) than did single-breed dogs (Est. = −0.303; SE = 0.009; p = <0.001) (S1 Table in S1 File). Under the same condition (with all other variables fixed), the mean Fear behavior score of the size variable reference group of smaller dogs (<20 lbs/~9kgs) was higher than that of all larger dog groups (weight >20lbs/~9kgs), that is, on average smaller dogs were more fearful (S1 Table in S1 File). Dogs classified by owners as companion/pet dogs had a Fear score average higher than service dogs, assistance/therapy dogs, and dogs classified as "Other" (S1 Table in S1 File). Puppies were less fearful on average than dogs at all other life stages; male dogs less fearful than females (behavior scores were 0.106 less on average); and the mean Fear score of a neutered/spayed dog was 0.274 higher than an intact dog, on average (i.e., on average, intact dogs reportedly displayed fewer fear-related behaviors than those who were neutered).

**Attention.** In the named domain Attention (PC2), one enrollment year, 2022, revealed significant differences from the reference enrollment year 2020. Dogs who enrolled in 2022 had Attention-related behavior scores that were on average 0.023 higher than those of dogs who enrolled in 2020 (S1 Table in S1 File).

Mixed-breed dogs had a higher average reported Attention score than did single-breed dogs. Puppies reportedly displayed more attention-related behaviors than did dogs at other life stages; females needed less attention than males; smaller dogs (<20lbs/~9kgs) demanded more than larger dogs (>20lbs/~9kgs); and dogs that were spayed/neutered had lower average Attention scores than the intact reference group. Primary area of residency was also associated with dogs' Attention-related behavior, with owners of dogs living in the Midwest reporting the highest average Attention score.

**Aggression.** In the domain including Aggression-related behaviors (PC3), there was a significant difference between average behavior scores among dogs who enrolled in 2023 compared with those that enrolled in 2020; 2023 enrollees' average Aggression behavior scores were 0.090 lower than 2020 enrollees' scores (S1 Table in S1 File).

Additionally, mixed breed dogs had 0.151 higher scores on average in the Aggression domain than single breed dogs (Est. = −0.151; SE = 0.009; p = <0.001). Puppies were reportedly less aggressive compared to dogs at all other life stages. Male dogs had Aggression scores on average 0.051 higher than females, and dogs that were neutered had reportedly higher Aggression scores on average compared to intact dogs. As in the Fear and Attention domains, smaller dogs were reportedly more aggressive than larger dogs. Region of residence was also associated with Aggression behavior scores. Compared to dogs living in the Midwest, owners of dogs living in the Northeast and the West were reportedly less aggressive (mean Aggression scores 0.031 lower in the Northeast and 0.05 lower in the West than the Midwest). Dogs classified by their owners as service dogs and therapy/assistance dogs displayed less aggression than companion animals/pets.

**Trainability.** Time (enrollment year) was impactful when it came to dogs' scores in our Trainability domain (PC4). Dogs enrolled in all three years after 2020 (2021–2023) had a lower average score in this domain than dogs enrolled in 2020; however, the mean difference between the Trainability score in 2020 enrollees and 2023 enrollees was the smallest among any of the years (S1 Table in S1 File).

Regarding other variables of interest, puppies had reportedly on average lower trainability scores than young and mature adult dogs, but were more trainable than senior dogs (S1 Table in S1 File). Females were more trainable than males with a 0.106 higher average behavior score, and dogs that were spayed/neutered were reportedly less trainable on average than those who were intact. Smaller dogs had lower trainability scores on average than dogs in other weight classes; dogs reported by owners to be living in the Northeast and in the South were less trainable than dogs in the reference group in the Midwest; and, dogs classified by owners as "obedience" and "other" had higher trainability scores than companion animals/pets.

## Discussion

Using the C-BARQ to measure individual dogs' scores in various behavioral domains, we established baseline behavioral characteristics for companion dogs enrolled in the DAP over four enrollment years. In exploring these data further, we find that while there are some changes in average scores of certain behavioral characteristics based on year of enrollment (2020–2023), greater variability in behavior appears to correlate with additional non-behavioral characteristics of interest.

We were especially interested in evaluating whether enrollment during specific years influenced reported behavior because our data were collected during a period in which a large proportion of U.S. households experienced disruption, starting with the beginning of the COVID-19 pandemic and ending in 2023 when many daily routines and pre-pandemic activities had been reestablished. Generally speaking, our results do not provide evidence of consistent reported behavior change trends in one direction or another. However, several findings imply continued investigation would be worthwhile. For example, where other behaviors fluctuated, we see no significant change in score averages in our named Fear domain over the study period, which supports the interpretation of some fear-related behaviors as part of a relatively intrinsic characteristic/personality feature in dogs [40].

Another example warranting further investigation involves our Aggression domain. We observed a significant decrease in average Aggression domain scores between dogs enrolled in 2020 and those enrolled in 2023, though not as a trend through those years (i.e., no significant difference between 2020 and 2021 enrollees, or 2020 and 2022 enrollees). While we cannot know from these data if changes in the household influenced owners' reports of aggression-related behaviors, one possible explanation could be reduced stress in the dogs' environments and more opportunities for social engagement in 2023 compared to at the height of the pandemic in 2020.

Finally, and most interestingly in regard to differences between enrollment years, dogs entering the study in all three years after 2020 (2021–2023) had lower average scores in Trainability than dogs that entered the study in 2020, regardless of life stage at time of entry. Again, while we cannot know the specific drivers of these differences, it is important to reiterate that the dogs' behavioral measures are as reported by owners, and it is possible that during the pandemic training proved to be challenging to owners (and dogs) for a host of reasons — e.g., people getting new dogs, people getting dogs who'd never had them before, routine and work shifts that impacted ability to train, etc. And, given that the mean difference between the Trainability score in 2020 and in 2023 was the smallest, is perhaps an indicator that the people/dogs were in a better position to train/be trained as they emerged from the pandemic.

Beyond enrollment year-based differences, we examined the relationships between average behavior scores in the various domains and multiple additional characteristics of interest. While we cannot know all of the environmental factors or potential confounding variables that may have influenced dogs' behavior or how it was recorded at the time of survey which may render some results spurious (including potential biases related to the owners' circumstances, relationship history with the dog, and inconsistency in interpretations of behaviors), analyses of these relationships nevertheless generated several notable findings.

Overall, mixed-breed dogs scored higher on average than single-breed dogs in the Fear, Attention, and Aggression domains, though not Trainability. Given that mixed-breed dogs more often than single-breed dogs are sourced from shelters and rescues where living conditions are stressful, it is possible that previous environmental trauma impacted dogs' reported behaviors in these realms [21,41,42]. Unsurprisingly, dogs whose primary activities as reported by owners included service and therapy had an overall lower mean score in the Aggression domain than did the reference group of companion/pet dogs. As service and therapy dogs undertake extensive training to assist people and work in highly human-centric environments, aggression would not be a characteristic well-tolerated for such tasks. In a similarly logical vein, dogs that reportedly participated in obedience activities scored higher overall on Trainability than did the companion/pet dog reference group.

In terms of dog life stage, puppies in each enrollment year appear to act very much the way they may be expected to, given their developmental needs. As a reference age group, compared to dogs at every other life stage (young adult, mature adult, senior), puppies' overall average reported behavior scores indicate they demand more attention, are less fearful and aggressive, but not as trainable – the latter of which is perhaps somewhat surprising given that they are primed for human interaction at early ages [43]. However, the difference in mean Trainability score is smallest between puppies and seniors, which could point to interesting implications for cognitive and behavioral responses of older dogs. Regarding the relationship between dog size and our four primary behavior domains, the data are not especially encouraging for small dog lovers. Small dogs (< 20lbs/44 kg) had higher average Fear, Aggression, and Attention scores and lower Trainability scores compared to dogs in each weight class >20lbs/44kgs. Here too, however, it is important to consider that data are subject to owners' interpretations, and characteristics such as trainability with small dogs could be related to the fact that smaller dogs can be more easily moved, restrained, picked-up, etc. than their larger counterparts, resulting in them simply being "handled" when displaying undesirable behaviors rather than being trained to act differently.

Differences in Aggression, Fear, and Trainability scores between males and females seen in our data generally align with some previous reports [44] (e.g., females are less bold and more trainable), though other studies have found that the overall effect of sex has little significance on much behavioral variation [45,46]. These latter findings may suggest sex-based or "gendered" perceptions of personality traits could contribute to reporting bias when dogs received their raw domain scores.

Residual confounding variables may likewise impact the real-world relevance of our results related to region of residence and spay/neuter status. Still, and despite the small number of gonadectomized individuals in the sample, it is worth remarking on the fact that intact dogs in our sample had lower average Aggression scores compared to the gonadectomized reference group. This finding aligns with recent work showing that de-sexing – commonly practiced as a means of behavior modification, particularly to reduce aggression – does not always result in animals with a more mild temperament, and may sometimes even have the opposite effect [47–49].

Finally, we did not find evidence that insured dogs had behavioral domain scores that were different from uninsured dogs, a result that prompts questions about the relationship between behavioral outcomes and physical health treatment opportunities and access to veterinary care.

## Conclusions

In this study we establish behavioral profiles based on owner-reported data for 47,444 dogs enrolled in the DAP, including enrollment year averages across 14 C-BARQ behavior domains. In our analyses of behavior and differences associated with enrollment year among four composite behavior domains — Fear, Attention/Excitability, Aggression, and Trainability — we find that year of enrollment was most relevant to Trainability-related behaviors. In all three enrollment years after 2020 (2021–2023) dogs had lower average reported Trainability scores than dogs that entered the study in 2020. Additional characteristics, including breed, life stage, sex, spay/neuter status, size, primary residence, and primary activities were associated with behavioral scores in all four domains. These results provide a critical starting point for continued consideration of behavior in a large-scale, longitudinal study of companion dogs, which will ultimately be necessary to inform investigations of relationships between behavior and physical health outcomes and cognitive changes as dogs age.

## Supporting information

**S1 File.** S1 Table. Results of regression analyses examining the relationships between variables of interest and mean behavior scores for dogs in four composite behavior domains: PC1 – Fear; PC2 – Attention; PC3 – Aggression; PC4 – Trainability. Dog Aging Project, 2020–2023. S2 Table. Results of initial regression models including interaction terms

examining the relationships between variables of interest and mean behavior scores for dogs in four composite behavior domains: PC1 – Fear; PC2 – Attention; PC3 – Aggression; PC4 – Trainability. Dog Aging Project, 2020–2023.
(PDF)

## Acknowledgments

We would like to thank Dr. James Serpell and the University of Pennsylvania for allowing us to include the shortened C-BARQ questionnaire in the Dog Aging Project. The authors also thank Dog Aging Project participants, their dogs, and community veterinarians for their important contributions. The content is solely the responsibility of the authors and does not necessarily represent the official views of the National Institutes of Health.

Members of the Dog Aging Project Consortium: Drs. Fitzpatrick and Ruple and the following authors of this report: Joshua M. Akey, Princeton, NJ; Brooke Benton, Seattle, WA; Elhanan Borenstein, Tel Aviv, Israel; Marta G. Castelhano, Ithaca, NY; Amanda E. Coleman, Athens, GA; Kate E. Creevy, College Station, TX; Matthew D. Dunbar, Seattle, WA; Virginia R. Fajt, College Station, TX; Jessica M. Hoffman, Augusta, GA; Erica C. Jonlin, Seattle, WA; Matt Kaeberlein, Seattle, WA; Elinor K. Karlsson, Worcester, MA; Kathleen F. Kerr, Seattle, WA; Jing Ma, Seattle, WA; Evan L. MacLean, Tucson, AZ; Daniel E. L. Promislow, Boston, MA; Stephen M. Schwartz, Seattle, WA; Sandi Shrager, Seattle, WA; Noah Snyder-Mackler, Tempe, AZ; M. Katherine Tolbert, College Station, TX; Silvan R. Urfer, Seattle, WA; and Benjamin S. Wilfond, Seattle, WA.

## Author contributions

**Conceptualization:** Yuhuan Li, Courtney Sexton, Annette Fitzpatrick, Audrey Ruple.

**Formal analysis:** Yuhuan Li.

**Investigation:** Yuhuan Li, Courtney Sexton, Audrey Ruple.

**Methodology:** Yuhuan Li, Annette Fitzpatrick.

**Supervision:** Annette Fitzpatrick, Audrey Ruple.

**Writing – original draft:** Yuhuan Li, Courtney Sexton.

**Writing – review & editing:** Yuhuan Li, Courtney Sexton, Annette Fitzpatrick, Audrey Ruple.

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
