## [Decision Letter · Decision Letter 0]

15 Apr 2025

PONE-D-25-10356An analysis of behavioral characteristics and enrollment year variability in more than 47,000 dogs entering the Dog Aging Project from 2020 to 2023PLOS ONE

Dear Dr. Sexton,

Thank you for submitting your manuscript to PLOS ONE. After careful consideration, we feel that it has merit but does not fully meet PLOS ONE’s publication criteria as it currently stands. Therefore, we invite you to submit a revised version of the manuscript that addresses the points raised during the review process.

We look forward to receiving your revised manuscript.

Kind regards,

Cord M. Brundage, D.V.M., Ph.D.

Academic Editor

PLOS ONE

2. If any supporting files for review show as item type ‘other’ please change to item type ‘supporting info’ as the reviewer does not have access to these ’other’ files.

3. Thank you for stating the following financial disclosure: [This research is based on publicly available data collected by the Dog Aging Project, under U19 grant AG057377 (PI: Daniel Promislow) from the National Institute on Aging, a part of the National Institutes of Health, and by additional grants and private donations, including generous support from the Glenn Foundation for Medical Research, the Tiny Foundation Fund at Myriad Canada, and the WoodNext Foundation.]. 

4. Thank you for uploading your study's underlying data set. Unfortunately, the repository you have noted in your Data Availability statement does not qualify as an acceptable data repository according to PLOS's standards.

5. One of the noted authors is a group or consortium [DAP Consortium]. In addition to naming the author group, please list the individual authors and affiliations within this group in the acknowledgments section of your manuscript. Please also indicate clearly a lead author for this group along with a contact email address.

Additional Editor Comments:

Please make sure that you address each of the reviewer comments with specific changes in the manuscript or provide a clear justification why those a change cannot be made. Any effort to indicate and explain the revisions that are made is greatly appreciated.

Reviewers' comments:

Reviewer's Responses to Questions

**Comments to the Author**

1. Is the manuscript technically sound, and do the data support the conclusions?

Reviewer #1: No

Reviewer #2: Yes

Reviewer #3: Yes

2. Has the statistical analysis been performed appropriately and rigorously? 

Reviewer #1: I Don't Know

Reviewer #2: Yes

Reviewer #3: I Don't Know

3. Have the authors made all data underlying the findings in their manuscript fully available?

Reviewer #1: No

Reviewer #2: No

Reviewer #3: Yes

4. Is the manuscript presented in an intelligible fashion and written in standard English?

Reviewer #1: Yes

Reviewer #2: Yes

Reviewer #3: Yes

5. Review Comments to the Author

Reviewer #1: The authors investigate changes in behavioral characteristics across 4 years of enrolled dogs in a large survey dataset. While the dataset itself is undoubtedly exciting and fruitful given its scale and scope, this manuscript ultimately falls short conceptually and methodologically. I cannot recommend it as suitable for publication at PLOS One.

My major concerns are the following:

1. Lack of clarity or compelling evidence for main claim: The central goal proposed by the authors is to create “baseline” behavioral measures for dog behavioral, but the authors do not clearly define what is meant by this baseline. Is this a baseline meant for future comparisons at the individual level? If so, I was not convinced that a separate manuscript needs to be published if there is no comparison group to each individual’s baseline presented. Or, is this meant to present group or sub-group level averages? The language used in the introduction and conclusion implies the baseline is important for tracking behavioral change, but the results presented here focus on group covariates. While I absolutely agree that understanding individuals’ baselines will be critical for future work, I was confused as to the goal of presenting them in isolation.

2. Context surrounding COVID-19 is critically underdeveloped: It seemed to me that the authors could not decide their position on how to approach potential variability introduced by data collection during the COVID 19 pandemic. The paper does not offer compelling analysis or consideration of its effects. For example, no consideration is given to whether the age at time of enrollment interacts with pandemic-related household disruptions, the age of the owner, the location of the owner, etc. A dog enrolled as a puppy in 2020 would have experienced a very different environment than a mature adult enrolled in the same year. These cohort effects deserve more thoughtful treatment, especially since the study hinges on comparisons across enrollment years.

3. Statistical methods are insufficiently described. The manuscript does not provide enough detail about the PCA or regression models used. No assessment of model fit and many effect sizes are left out, raising concerns about negligible effects. The authors do not describe the naming of their components or internal consistency of components. The PCA reduces 14 C-BARQ domains to four components, but the logic behind this reduction is only minimally described. Authors in the results section often report that scores are “higher” without any supporting statistical results.

4. Limited engagement with the literature and unclear contribution. The manuscript does not convincingly situate itself within existing behavioral literature. Many of the findings such as differences in behavior by size or breed are well-documented yet the authors do not identify the specific gap that their study fills. Simply stating that this is a “large” sample is not sufficient. The paper would benefit from a more focused articulation of its novelty and relevance.

5. No discussion or argument for CBARQ construct validity. The authors also describe behavior as a “simple” measure of emotional state, which I believe is a gross oversimplification of the nuance of animal behavior.

6. Claims about relevance to human health are underdeveloped. The manuscript repeatedly refers to the importance of dogs as sentinels for human health and aging, but this idea is not explored in a meaningful way.

Summary: While the Dog Aging Project dataset has significant potential, this manuscript does not offer a conceptually or methodologically sound analysis of that data. The framing and analytical strategy lacks clarity, and the conclusions are vague. For these reasons, I do not believe the manuscript meets PLOS ONE’s standards for publication in its current form. I encourage the authors to reconsider the framing of their research question and substantially revise their analysis and interpretive approach to understand fully the interplay of behavior and environment.

Reviewer #2: This study is clearly described with a clear rationale, and the statistical analysis is explained in detail. My comments are limited to some very minor issues with the wording and some concerns about the discussion, which I think does not sufficiently explore the possible limitations of an owner-described dataset.

Minor issues with the wording:

line 139 - 'representative' - should this be 'represented'? - I didn't understand this.

line 155 - I found the frequent switching between present and past tense in this section of the methods very jolting to read. Can you reword to change tense less frequently?

line 387 - use of the word 'demure' - I know 'demure' was having a moment on TikTok at the time this paper was probably being written, but the definition of the word is 'reserved, modest and shy' - I don't think a dog can be modest and I think the use of this word is anthropomorphic - suggest you use 'calmer' or similar instead.

Concerns about the discussion - I am not familiar with the questionnaire used, so it would have been helpful to include one or two sample questions or a brief description of the sort of information that the owners were expected to provide. This study relies on owner-reported observations of canine behaviour, but we know from other research that owners are often terrible at correctly interpreting their dog's behaviour and misunderstand canine body language. You say that the shortened version of the questionnaire was validated by researchers, but then you seem to assume that the owners' reported behavioural observations are reliable without either explaining why you think this or what uncertainty this might introduce if they are not interpreting canine body language successfully. Given that many owners e.g. confuse fear and aggression and misinterpret tail movements, ear position, etc, this seems to me like a possible major weakness in the study.

Similarly, the discussion assumes that the judgments made by the owners are all impartial and comparable. But different types of people may choose different types of dog. Small dogs might have lower Trainability scores because they are less trainable, or perhaps their owners perceive them as less trainable or put less effort into training them than is the case for larger dog owners. Similarly, owners who do obedience with their dogs might actually have dogs with higher trainability, or might rate trainability more highly because they care enough about it to get involved with obedience! I think there needs to be at least some acknowledgement that some of this data is potentially subjective because you cannot control for the different biases that owners of different categories of dog might bring to their assessments.

I suggest that the authors might like to look at the discussion sections of some of the owner-reported canine behavioural studies conducted by Rowena Packer (not me!) and various colleagues, because I think these do a good job of unpicking the biases that may come from owner-reported data.

In summary, I would like to see a discussion that is expanded to consider and address possible sources of bias in the data in more detail.

Reviewer #3: The article is easy to read, but hard to interpret. I am not sure what the implications of this study and its findings are, or what problem was aimed to solve/knowledge gap was to be filled. I think the main message of the study is (/should be) that you can use the C-BARQ to identify 4 main personality domains and then which factors influence each of those domains. And DAP data was used for this. However, the focus on the article seems to be on aging and difference between years and the DAP, while that seems to be of minimal importance to what was actually done. Moreover, I don’t think the authors looked at interactions between influencing factors, while I expect there to be some. I am not an expert in the topic of statistic though, so I do not dare say whether the analysis has been performed appropriately and rigorously other than that. Nonetheless, I think the results are interesting, but I would frame the article differently to make the importance and the ‘coolness’ of your results more evident. I have provided feedback in more detail below.

Abstract: It is not clear what the main rationale behind the study was, nor what your results are saying/meaning. The first part is fine, but then I got confused at the sentence ‘when considering dogs as sentinels of human health and aging’ (42-43). What do you mean with ‘sentinels’? Indicators? Or do you mean more as using dogs as models for human health? I am not a native speaker, but even a google search gave me no logic translation (‘ a soldier standing guard’). Maybe change that word to make it clearer. The second part of the sentence indicates that you probably mean it in the way that dogs can be models for humans, but then nothing else in the abstract points in the direction of this study is about using dogs as models for humans.. What does understanding dog behaviour being important for interactions and their health etc have to do with dogs being used as models for humans? And, if you do this study to say something about humans, why do the humans not come back in the last part of the abstract? Seems to me like you are focusing this project on understanding the baseline characteristics of the dogs that subscribed to the DAP, which you could potentially use later for comparison with humans/as models for humans, but that was not the main rationale here. If it was, it should be clearer. It is also unclear to me what your results implicate from reading your abstract alone. You write that time has the highest influence on trainability. In what way? Older/younger dogs becoming more/less trainable? And what does it mean that several variables are associated with mean behavioural scores? All those factors influence all behaviours? In what way? The abstract should be a summary of your article, but after reading your abstract I still know barely anything, so I would really rewrite that.

81-82 Here you again mention the humans (more as a side mention here in the intro then in the abstract), but it is still unclear to me why you could use dogs as models to humans. Maybe you can include some literature on how they are alike?

84-89 this feels like an unfinished, stand-alone paragraph, which really stops the reading flow. Maybe add some studies that used this tool and what was previously found using this tool, to link it better to the next paragraph and improve the flow in the writing. Maybe you can also link it to be previous paragraph with using a sentence like ‘A dog’s baseline behavior can be assessed with a questionnaire such as the Canine Behavioral Assessment and Research Questionnaire (C-BARQ), which is….’. However, that would mostly work if the part on understanding humans wouldn’t be in between.

93 It would be nice to get some background on the HLES, even if it is an added sentence like ‘, which dog owners fill in once they subscribe their dog for the DAP’.

101-106 On your second aim, the difference between years, there is now not really anything in the introduction, so I wonder what the relevance of this aim is? What are you trying to answer/solve her? Also, there is no aiming on using dogs as models for humans etc., so if possible I would just leave it out of your intro and abstract. If you need to put it in because of funding or similar, elaborate more on it and also put it in your aims.

136 When you say the C-BARQ produces domains, you mean each dog gets a score for each of the 14 domains, or how do I see this?

155-160 Did you also look for interactions? I can imagine that if a certain area shows elevated scores, but a certain breed/size as well that there might be an interaction (many dogs of that kind in that area)

163 Was the mortality data previously described? Where?

173-174 what test did you use to compare the means between years?

186 delete the , between dogs and the corresponding percentage

201-205 where these differences significant? Can you add a test statistic and p-value?

242-248 Can you include test statistics and p-values?

254 I would start more general (as with fear), not directly with 2022.

259-261 It would be nice to have an reiteration of what an high attention score means. Showing more attention seeking behaviours?

274-283 Add test statistics and p-values

287-291 You put this as your main result in your abstract, but also here I am unsure what this means/implies. Seems to me like a coincidence since being enrolled isn’t really influencing anything for the dog? Or does this represent age of the dog..?

293-300 Add test statistics and p-values

311-315 If your aim was to see if there were behaviour change trends in dogs, due to the Covid pandemic this should have been more evident in your introduction (talk about behaviour changes and disruptions and how earlier studies have found effects of sudden differences in contexts on the behaviour of animals, and specifically on the behaviour of dogs (eg more separation anxiety etc) and build a case from there) and I would put it as an aim. It is a cool angle and one I would really use more.

312 change the second ‘during’ to ‘in’

331-333 Here is the first time I really understand what you mean with this result and this should really be clearer before (and makes more since in the light of the aim to find out differences due to covid years)

Generally discussion: I always tell my students to compare your own results with literature and then really include also what they did in that other study so it is also clear for the reader how these compare. You have a very minimal amount of other studies mentioned in your discussion and I think the story would be stronger if you would do so. There must be other studies looking into what factors influence fear, aggression, attention seeking etc., there must be other studies looking into sex or size differences etc. Did they find the same thing as you? Why (not)?

Conclusion: the results are unclear again and seem to be copied from the abstract (or the other way around). The last part is strong and the abstract would also improve from an addition like that (but not identical!)

6. PLOS authors have the option to publish the peer review history of their article (what does this mean? ). If published, this will include your full peer review and any attached files.

**Do you want your identity to be public for this peer review?** For information about this choice, including consent withdrawal, please see our Privacy Policy .

Reviewer #1: No

Reviewer #2: No

Reviewer #3: No

---

## [Author Response · Author response to Decision Letter 1]

17 Jun 2025

An analysis of behavioral characteristics and enrollment year variability in more than 47,000 dogs entering the Dog Aging Project from 2020 to 2023

PLOS ONE

Response to Reviewers:

Reviewer #1: The authors investigate changes in behavioral characteristics across 4 years of enrolled dogs in a large survey dataset. While the dataset itself is undoubtedly exciting and fruitful given its scale and scope, this manuscript ultimately falls short conceptually and methodologically. I cannot recommend it as suitable for publication at PLOS One.

My major concerns are the following:

1. Lack of clarity or compelling evidence for main claim: The central goal proposed by the authors is to create “baseline” behavioral measures for dog behavioral, but the authors do not clearly define what is meant by this baseline. Is this a baseline meant for future comparisons at the individual level? If so, I was not convinced that a separate manuscript needs to be published if there is no comparison group to each individual’s baseline presented. Or, is this meant to present group or sub-group level averages? The language used in the introduction and conclusion implies the baseline is important for tracking behavioral change, but the results presented here focus on group covariates. While I absolutely agree that understanding individuals’ baselines will be critical for future work, I was confused as to the goal of presenting them in isolation.

We appreciate this concern. At the individual level, the HLES serves as a baseline for each enrolled dog, a measure that can be compared against subsequent annual follow-up surveys submitted by participants. Our goal in this manuscript was to evaluate an overall baseline describing cohort entry year averages. This is the first group-level publication of behavioral characteristics of a long-term longitudinal study in which dogs will continue to age. With these group-level measures established, we’ll be able to track changes for the whole population over time, in addition to individual changes.

We have made an effort to clarify in the Introduction the importance of the group-level analysis (Lines 96-99; 130-134).

2. Context surrounding COVID-19 is critically underdeveloped: It seemed to me that the authors could not decide their position on how to approach potential variability introduced by data collection during the COVID 19 pandemic. The paper does not offer compelling analysis or consideration of its effects. For example, no consideration is given to whether the age at time of enrollment interacts with pandemic-related household disruptions, the age of the owner, the location of the owner, etc. A dog enrolled as a puppy in 2020 would have experienced a very different environment than a mature adult enrolled in the same year. These cohort effects deserve more thoughtful treatment, especially since the study hinges on comparisons across enrollment years.

Thank you for this comment. We have included additional context regarding the COVID-19 pandemic and its relevance to our study (Line 50; Line 77-81; Line 130; Line 205; Line 349-353).

Interaction terms were included in the model and not found to be statistically significant and so were not included in the final statistical model. Should the editors find it prudent to include these initial tests as supplement, we would be happy to provide.

3. Statistical methods are insufficiently described. The manuscript does not provide enough detail about the PCA or regression models used. No assessment of model fit and many effect sizes are left out, raising concerns about negligible effects. The authors do not describe the naming of their components or internal consistency of components. The PCA reduces 14 C-BARQ domains to four components, but the logic behind this reduction is only minimally described. Authors in the results section often report that scores are “higher” without any supporting statistical results.

Detailed descriptions of the PCA pre-analysis and PCA are included in the Results section, including domain loadings and clustering for named areas. Should the editors determine it would be more aligned with the journal standards to include this in the Methods section, we would be happy to make that adjustment.

Re: additional statistical measures, given the large number of score valuations presented in the Results section, we opted to report full results including all ns, estimates, SEs, and p-values in Table S1. It has been brought to our attention that this table was not available to reviewers in our initial submission.

4. Limited engagement with the literature and unclear contribution. The manuscript does not convincingly situate itself within existing behavioral literature. Many of the findings such as differences in behavior by size or breed are well-documented yet the authors do not identify the specific gap that their study fills. Simply stating that this is a “large” sample is not sufficient. The paper would benefit from a more focused articulation of its novelty and relevance.

We understand this concern, but the large sample size is the key component of our investigation. Our primary aim in this manuscript is to establish group baselines for a specific element of a large dataset that can be used in future analyses with the dataset. Additionally, given that a substantial amount of prior work/existing behavioral literature focuses on single breeds or draws from single institutions/sources, we did not feel it relevant to include additional background on these studies at this time.

5. No discussion or argument for CBARQ construct validity. The authors also describe behavior as a “simple” measure of emotional state, which I believe is a gross oversimplification of the nuance of animal behavior.

We have included additional information and citations regarding the validation and use of C-BARQ in various behavioral studies.

We certainly did not mean to imply that behavior is not complexed or nuanced and have removed this term.

6. Claims about relevance to human health are underdeveloped. The manuscript repeatedly refers to the importance of dogs as sentinels for human health and aging, but this idea is not explored in a meaningful way.

One of the promising directions of the Dog Aging Project is the potential to use the massive amounts of data collected about dogs who live in the same social and physical environments with their people in a translational manner in terms of investigating similar external risk factors and how they may impact health and aging in both dogs and people. We have clarified this relationship in the manuscript and included references (Lines 93-96).

Summary: While the Dog Aging Project dataset has significant potential, this manuscript does not offer a conceptually or methodologically sound analysis of that data. The framing and analytical strategy lacks clarity, and the conclusions are vague. For these reasons, I do not believe the manuscript meets PLOS ONE’s standards for publication in its current form. I encourage the authors to reconsider the framing of their research question and substantially revise their analysis and interpretive approach to understand fully the interplay of behavior and environment.

Thank you for your thoughtful insights. We have taken these and other reviewers’ suggestions into consideration and have made considerable revisions to the manuscript to clarify its purpose and findings.

Reviewer #2: This study is clearly described with a clear rationale, and the statistical analysis is explained in detail. My comments are limited to some very minor issues with the wording and some concerns about the discussion, which I think does not sufficiently explore the possible limitations of an owner-described dataset.

Thank you for taking the time to provide your expert review of our manuscript; we appreciate your feedback and have addressed the issues raised.

Minor issues with the wording:

line 139 - 'representative' - should this be 'represented'? - I didn't understand this.

Thank you for catching this error; it has been corrected.

line 155 - I found the frequent switching between present and past tense in this section of the methods very jolting to read. Can you reword to change tense less frequently?

We have made this adjustment.

line 387 - use of the word 'demure' - I know 'demure' was having a moment on TikTok at the time this paper was probably being written, but the definition of the word is 'reserved, modest and shy' - I don't think a dog can be modest and I think the use of this word is anthropomorphic - suggest you use 'calmer' or similar instead.

We have made this adjustment.

Concerns about the discussion - I am not familiar with the questionnaire used, so it would have been helpful to include one or two sample questions or a brief description of the sort of information that the owners were expected to provide. This study relies on owner-reported observations of canine behaviour, but we know from other research that owners are often terrible at correctly interpreting their dog's behaviour and misunderstand canine body language. You say that the shortened version of the questionnaire was validated by researchers, but then you seem to assume that the owners' reported behavioural observations are reliable without either explaining why you think this or what uncertainty this might introduce if they are not interpreting canine body language successfully. Given that many owners e.g. confuse fear and aggression and misinterpret tail movements, ear position, etc, this seems to me like a possible major weakness in the study.

We recognize that owner-reported data can be more or less reliable depending on the context. We have addressed this concern in the introduction and included additional information and citations regarding the validation and use of C-BARQ in various behavioral studies (Lines 106-111). Notably, the guide dog organization Canine Companions uses C-BARQ for generating reports from at-home puppy raisers of future service dogs.

Similarly, the discussion assumes that the judgments made by the owners are all impartial and comparable. But different types of people may choose different types of dog. Small dogs might have lower Trainability scores because they are less trainable, or perhaps their owners perceive them as less trainable or put less effort into training them than is the case for larger dog owners. Similarly, owners who do obedience with their dogs might actually have dogs with higher trainability, or might rate trainability more highly because they care enough about it to get involved with obedience! I think there needs to be at least some acknowledgement that some of this data is potentially subjective because you cannot control for the different biases that owners of different categories of dog might bring to their assessments.

We recognize that owner-reported data may introduce certain biases and have been sure to make clear that we are analyzing and interpreting owner-reported data throughout the manuscript. Given that in this report we are primarily investigating cohort year trends (as opposed to individual differences) across behavioral domains using a tool validated specifically for use in owner/non-experimenter-reported data (CBARQ) with a very large sample size (n=47,444), we do not believe these potential biases substantially impact this particular analysis, though are nevertheless interesting points for discussion and we have expanded on them in that section as suggested.

I suggest that the authors might like to look at the discussion sections of some of the owner-reported canine behavioural studies conducted by Rowena Packer (not me!) and various colleagues, because I think these do a good job of unpicking the biases that may come from owner-reported data.

Thank you for suggesting these references – very interesting work in an important arena!

In summary, I would like to see a discussion that is expanded to consider and address possible sources of bias in the data in more detail.

The discussion of these biases and potential interpretations related to them is present in our Discussion, and we have expanded the on potential relevance/import in the section (Lines 381-386; 411-415), per your recommendation.

Reviewer #3: The article is easy to read, but hard to interpret. I am not sure what the implications of this study and its findings are, or what problem was aimed to solve/knowledge gap was to be filled. I think the main message of the study is (/should be) that you can use the C-BARQ to identify 4 main personality domains and then which factors influence each of those domains. And DAP data was used for this. However, the focus on the article seems to be on aging and difference between years and the DAP, while that seems to be of minimal importance to what was actually done. Moreover, I don’t think the authors looked at interactions between influencing factors, while I expect there to be some. I am not an expert in the topic of statistic though, so I do not dare say whether the analysis has been performed appropriately and rigorously other than that. Nonetheless, I think the results are interesting, but I would frame the article differently to make the importance and the ‘coolness’ of your results more evident. I have provided feedback in more detail below.

Thank you for taking the time to provide your expert review of our manuscript; we appreciate your feedback and have addressed the issues raised.

Our defined aims for this study are 1) to describe behavioral data for specific characteristics for cohorts of DAP dogs at their time of enrollment; and 2) to examine behavioral data among cohorts of dogs enrolled each year over the lifetime of the project for similarities and differences and investigate additional variables that may impact observed differences. (Lines 126-131)

Abstract: It is not clear what the main rationale behind the study was, nor what your results are saying/meaning. The first part is fine, but then I got confused at the sentence ‘when considering dogs as sentinels of human health and aging’ (42-43). What do you mean with ‘sentinels’? Indicators? Or do you mean more as using dogs as models for human health? I am not a native speaker, but even a google search gave me no logic translation (‘ a soldier standing guard’). Maybe change that word to make it clearer. The second part of the sentence indicates that you probably mean it in the way that dogs can be models for humans, but then nothing else in the abstract points in the direction of this study is about using dogs as models for humans.. What does understanding dog behaviour being important for interactions and their health etc have to do with dogs being used as models for humans? And, if you do this study to say something about humans, why do the humans not come back in the last part of the abstract? Seems to me like you are focusing this project on understanding the baseline characteristics of the dogs that subscribed to the DAP, which you could potentially use later for comparison with humans/as models for humans, but that was not the main rationale here. If it was, it should be clearer. It is also unclear to me what your results implicate from reading your abstract alone. You write that time has the highest influence on trainability. In what way? Older/younger dogs becoming more/less trainable? And what does it mean that several variables are associated with mean behavioural scores? All those factors influence all behaviours? In what way? The abstract should be a summary of your article, but after reading your abstract I still know barely anything, so I would really rewrite that.

We have edited our abstract to clarify our definition of dogs as sentinels, the aims of the study, and the summary of our findings.

81-82 Here you again mention the humans (more as a side mention here in the intro then in the abstract), but it is still unclear to me why you could use dogs as models to humans. Maybe you can include some literature on how they are alike?

We have clarified this relationship and included references (Lines 93-96).

84-89 this feels like an unfinished, stand-alone paragraph, which really stops

---

## [Decision Letter · Decision Letter 1]

8 Jul 2025

PONE-D-25-10356R1An analysis of behavioral characteristics and enrollment year variability in more than 47,000 dogs entering the Dog Aging Project from 2020 to 2023PLOS ONE

Dear Dr. Sexton,

Thank you for submitting your manuscript to PLOS ONE. After careful consideration, we feel that it has merit but does not fully meet PLOS ONE’s publication criteria as it currently stands. Therefore, we invite you to submit a revised version of the manuscript that addresses the points raised during the review process.

We look forward to receiving your revised manuscript.

Kind regards,

Cord M. Brundage, D.V.M., Ph.D.

Academic Editor

PLOS ONE

Additional Editor Comments (if provided):

Please make sure that you fully address these valid reviewer comments and concerns both in your response to reviewers and within the manuscript.

Reviewers' comments:

Reviewer's Responses to Questions

**Comments to the Author**

1. If the authors have adequately addressed your comments raised in a previous round of review and you feel that this manuscript is now acceptable for publication, you may indicate that here to bypass the “Comments to the Author” section, enter your conflict of interest statement in the “Confidential to Editor” section, and submit your "Accept" recommendation.

Reviewer #2: All comments have been addressed

Reviewer #4: All comments have been addressed

Reviewer #5: (No Response)

2. Is the manuscript technically sound, and do the data support the conclusions?

Reviewer #2: Yes

Reviewer #4: No

Reviewer #5: Yes

3. Has the statistical analysis been performed appropriately and rigorously? 

Reviewer #2: I Don't Know

Reviewer #4: No

Reviewer #5: I Don't Know

4. Have the authors made all data underlying the findings in their manuscript fully available?

Reviewer #2: Yes

Reviewer #4: No

Reviewer #5: Yes

5. Is the manuscript presented in an intelligible fashion and written in standard English?

Reviewer #2: Yes

Reviewer #4: Yes

Reviewer #5: Yes

6. Review Comments to the Author

Reviewer #2: The responses to the comments left by all three reviewers have considerably improved the paper's clarity and better articulated how it contributes to the literature. Looking at the other reviewers' comments, I think that the expertise of reviewer 1 in particular in statistics is much greater than mine, so I don't feel qualified to critique the statistical analysis in detail. My only remaining comment (and I apologise for not mentioning this before) is that the dogs' weights are measured and grouped in pounds. As far as I know, the USA is the only country that still uses imperial measurements in scientific publications. I'm British and familiar with both systems as I grew up with imperial, but if this paper is intended to be readily accessible to a global audience then I suggest adding kilogram equivalents to the weight bands you describe, so that international readers can appreciate these categories without having to deal with a system they have never used.

Reviewer #4: The structure of your study is very well-designed, and the research topic itself is highly valuable. However, when transforming this study into a publishable manuscript, it is important to note that online surveys often lack reliability, particularly in studies involving animals. For instance, when classifying the dogs, the information provided by the owners or breeders may be inaccurate or unverified. Especially in the 'purebred' category, it is unclear whether the dogs have official pedigrees or not. Additionally, certain physical traits—such as categorization by size (small vs. large breeds)—cannot be confirmed through visual inspection in an online setting. Considering that the manuscript is being submitted to a Q1-ranked journal, while the study topic is indeed promising, it would be more robust and scientifically sound if the data were collected in person, with photographic documentation and visual confirmation of the animals. Such an approach would enhance the credibility and accuracy of the findings.

Reviewer #5: Thank you to the authors for this well written study. I was invited as a reviewer to assess the revision and response to reviewers. The manuscript in well organized, well written and highlights an interesting study using the information from the Dog Aging Project.

Overall, I believe there is merit to this study being published, however there are edits I suggest to improve the manuscript. I do highly disagree with the idea of connecting this study to human health – the argument is not well constructed and lacks supporting ideas. Additionally, the study presented in this manuscript is only on dog health, which cannot be comparable. I recommend removing all mention of connections to human health and instead focus on how the results from this study can benefit dog health.

All line numbers refer to the tracked change version, which I’ve included as an attached Word Document.

7. PLOS authors have the option to publish the peer review history of their article (what does this mean? ). If published, this will include your full peer review and any attached files.

**Do you want your identity to be public for this peer review?** For information about this choice, including consent withdrawal, please see our Privacy Policy .

Reviewer #2: No

Reviewer #4: **Yes: ** Dr. Ercan Mevliyaoğulları

Reviewer #5: No

---

## [Author Response · Author response to Decision Letter 2]

16 Jul 2025

PONE-D-25-10356R1

An analysis of behavioral characteristics and enrollment year variability in 47,444 dogs entering the Dog Aging Project from 2020 to 2023

PLOS ONE

Response to Reviewers:

Reviewer #2: The responses to the comments left by all three reviewers have considerably improved the paper's clarity and better articulated how it contributes to the literature. Looking at the other reviewers' comments, I think that the expertise of reviewer 1 in particular in statistics is much greater than mine, so I don't feel qualified to critique the statistical analysis in detail. My only remaining comment (and I apologise for not mentioning this before) is that the dogs' weights are measured and grouped in pounds. As far as I know, the USA is the only country that still uses imperial measurements in scientific publications. I'm British and familiar with both systems as I grew up with imperial, but if this paper is intended to be readily accessible to a global audience then I suggest adding kilogram equivalents to the weight bands you describe, so that international readers can appreciate these categories without having to deal with a system they have never used.

- Thank you for your additional time and consideration; we are glad to hear that our revisions have improved the manuscript in your view. Thanks also for this recommendation regarding measurement units. Because the original survey was available to dog owners living in the U.S., U.S. units were provided, and we thus reported them as such for consistency. We agree, however, that it is very silly that this system is still used and have updated the manuscript to include conversions. Thank you!

Reviewer #4: The structure of your study is very well-designed, and the research topic itself is highly valuable. However, when transforming this study into a publishable manuscript, it is important to note that online surveys often lack reliability, particularly in studies involving animals. For instance, when classifying the dogs, the information provided by the owners or breeders may be inaccurate or unverified. Especially in the 'purebred' category, it is unclear whether the dogs have official pedigrees or not. Additionally, certain physical traits—such as categorization by size (small vs. large breeds)—cannot be confirmed through visual inspection in an online setting. Considering that the manuscript is being submitted to a Q1-ranked journal, while the study topic is indeed promising, it would be more robust and scientifically sound if the data were collected in person, with photographic documentation and visual confirmation of the animals. Such an approach would enhance the credibility and accuracy of the findings.

- Thank you for your additional time and consideration. While we do recognize and note the inherent challenges of working with owner-reported data, and data validation is a core component of the Dog Aging Project. Dog Aging Project data (https://data.dogagingproject.org/Index) do not come from a single online survey, rather comprises a series of validated questionnaires (see McNulty et al. 2023 and Wilkins et al. 2024) completed by owners via a personalized portal, biospecimens, veterinary clinical records (which include size/weight), and additional datatypes for certain cohorts. These data have been used extensively in investigations across fields, with findings published widely and in highly ranked journals (see: https://dogagingproject.org/publications).

Regarding breed verification, we have a publication forthcoming (in press, Scientific Reports) in which we compared owner breed reporting to genetic panel results and found very high concordance for both single and mixed breed dogs.

Reviewer #5: Thank you to the authors for this well written study. I was invited as a reviewer to assess the revision and response to reviewers. The manuscript in well organized, well written and highlights an interesting study using the information from the Dog Aging Project.

Overall, I believe there is merit to this study being published, however there are edits I suggest to improve the manuscript. I do highly disagree with the idea of connecting this study to human health – the argument is not well constructed and lacks supporting ideas. Additionally, the study presented in this manuscript is only on dog health, which cannot be comparable. I recommend removing all mention of connections to human health and instead focus on how the results from this study can benefit dog health.

All line numbers refer to the tracked change version, which I’ve included as an attached Word Document.

- Thank you very much for taking the time to provide such thoughtful feedback. We have taken your comments and the suggestions of other reviewers to heart and have reconsidered our inclusion of the sentinel narrative (connections to human health) in this manuscript. We have likewise addressed your additional comments below.

Specific Comments

Title and abstract – It would be more appropriate to list exactly how many dogs were used in the study. ‘47,444 dogs’ is different from ‘more than 40,000 dogs’ and highlights the breath of the study and the number of enrolled dogs (which is a great amount!).

- We have made this change.

Abstract – From my understanding, the journals maximum word count for the abstract is 300 words, and the abstract is currently at 308. The authors should trim down to match the journals guidelines.

- We have revised the abstract and ensured it is compliant with word count.

L53 – I recommend removing mention of ‘human aging pattens’ since this was not analysed in this study (thus, it cannot be an aim of the study).

- We have made this change.

L54 and L83, and further instances throughout the manuscript – please use ‘COVID-19 lockdown’ vs ‘Covid lockdown’ (technically COVID itself it stands for ‘coronavirus disease 19’, and the authors should expand the acronym on first use in both the abstract and body of the manuscript).

- We have amended to the proper nomenclature.

L101 – It would be worthwhile to mention that the questions for the C-BARQ is measured on a 5-point scale.

- We have included mention of the scale.

L103 – It might be clearer for the reader if you direct them to Table 1 from here (since this is first mention of the 14 domains) but then remind them at L103 again.

- We have included a reference to Table 1 here.

Statistics – In regard to the authors response to reviewer 1, I’m of the opinion that the authors should include the initial tests (with interaction terms), in the methods (and subsequent results section). This is pertinent information that the reader should be made aware of.

- We have updated the Methods and Results sections to include the initial tests and equations and have provided the results from the interaction terms model in an additional supplementary table (Table S2).

L203, L226, and further instances – The journal uses SI units, and the authors should use kilograms here (in parathesis if it’s the case that DAP used lbs).

- Yes! Another reviewer just flagged that as well. The DAP surveys are in lbs. but we have updated to include the conversions in parentheses.

L215 – Please include citations for the packages since they are intellectual property.

- We have included these citations.

Results – When the authors present percentages, directly after, they should include the numerator/denominator in parentheses.

- Where percentages are given, we already indicate in the text the total number that the percentages are out of, and the numerical totals are present in Table 2 so we have opted to keep these reported as percentages.

Table 2 (and Table 3, Table 4) – I just want to mention that I really like how the authors set up this table – it’s very well organized and clear. I just recommend that the authors expand all abbreviations and define ‘n/year’. The captions should be understood without reference to the text.

- Thank you very much! And we have made these adjustments.

I’m not convinced that this study sheds light on the human health and aging. Especially since a major result in the study relates to ‘trainability’ (which would be inappropriate to connect to humans in this context). I would advise to the authors to remove all sentences that allude to the human aging process, since none of the results in this study could be related to human aging (since I would argue human aging is much more complex with many more factors than what was presented in this study). Additionally, the authors did not explore any external factors that could impact both humans and dogs alike, rendering the conclusion that the results could translate to insight into human aging incorrect. For the authors to make this conclusion, a study would investigate both humans and dogs alike simultaneously (of which this manuscript does not accomplish).

- We have removed this narrative thread throughout. Thank you.

---

## [Decision Letter · Decision Letter 2]

30 Jul 2025

An analysis of behavioral characteristics and enrollment year variability in 47,444 dogs entering the Dog Aging Project from 2020 to 2023

PONE-D-25-10356R2

Dear Dr. Sexton,

We’re pleased to inform you that your manuscript has been judged scientifically suitable for publication and will be formally accepted for publication once it meets all outstanding technical requirements.

Kind regards,

Cord M. Brundage, D.V.M., Ph.D.

Academic Editor

PLOS ONE

Reviewers' comments:

Reviewer's Responses to Questions

**Comments to the Author**

1. If the authors have adequately addressed your comments raised in a previous round of review and you feel that this manuscript is now acceptable for publication, you may indicate that here to bypass the “Comments to the Author” section, enter your conflict of interest statement in the “Confidential to Editor” section, and submit your "Accept" recommendation.

Reviewer #4: All comments have been addressed

Reviewer #5: (No Response)

2. Is the manuscript technically sound, and do the data support the conclusions?

Reviewer #4: Yes

Reviewer #5: Yes

3. Has the statistical analysis been performed appropriately and rigorously? 

Reviewer #4: Yes

Reviewer #5: Yes

4. Have the authors made all data underlying the findings in their manuscript fully available?

Reviewer #4: Yes

Reviewer #5: Yes

5. Is the manuscript presented in an intelligible fashion and written in standard English?

Reviewer #4: Yes

Reviewer #5: Yes

6. Review Comments to the Author

Reviewer #4: (No Response)

Reviewer #5: Thank you to the authors for their tremendous work in revising and editing their manuscript. In my opinion, the manuscript can now be accepted. I do have two small suggestions, but the authors can choose to incorporate at their discretion. the manuscript will not need to reviewed again.

(Line numbers are from the tracked changed version)

- L207 – I’m of the opinion that the authors should not include the models in the text of the manuscript. Either this should be explained in a paragraph form, or the models should be included as a supplementary, but the authors can choose at their discretion.

- Tables 3 and 4 – acronyms should be expanded on first use in tables.

Thank you for the opportunity to review your manuscript.

7. PLOS authors have the option to publish the peer review history of their article (what does this mean? ). If published, this will include your full peer review and any attached files.

**Do you want your identity to be public for this peer review?** For information about this choice, including consent withdrawal, please see our Privacy Policy .

Reviewer #4: No

Reviewer #5: No
